# Neuronal Capacity

**Pierre Baldi**
Department of Computer Science
University of California, Irvine
Irvine, CA 92697
pfbaldi@uci.edu

**Roman Vershynin**
Department of Mathematics
University of California, Irvine
Irvine, CA 92697
rvershyn@uci.edu

## Abstract

We define the capacity of a learning machine to be the logarithm of the number (or volume) of the functions it can implement. We review known results, and derive new results, estimating the capacity of several neuronal models: linear and polynomial threshold gates, linear and polynomial threshold gates with constrained weights (binary weights, positive weights), and ReLU neurons. We also derive some capacity estimates and bounds for fully recurrent networks, as well as feedforward networks.

## 1 Introduction

A basic framework for the study of learning (Figure 1) consists in having a target function $h$ that one wishes to learn and a class of functions or hypothesis $A$ that is available to the learner to implement or approximate $h$. The class $A$, for instance, could be all the functions that can be implemented by a given neural network architecture as the synaptic weights are varied. Obviously how well $h$ can be learnt critically depends on the class $A$ and thus it is natural to seek to define a notion of "capacity" for any class $A$. The goal of this paper is to define a notion of capacity and show how it can be computed, or approximated, in the case of several neural models. As a first step, in this paper we define the capacity of the class $A$ to be the logarithm base two of the size or volume of $A$:

$$C(A) = \log_2 |A| \tag{1}$$

This is also the number of bits that can be communicated, or stored, by selecting an element of $A$. Needless to say this notion of capacity is only a first step towards characterizing the capabilities of $A$ and which kinds of function it is capable of learning, a problem that has remained largely out of reach for neural architectures.

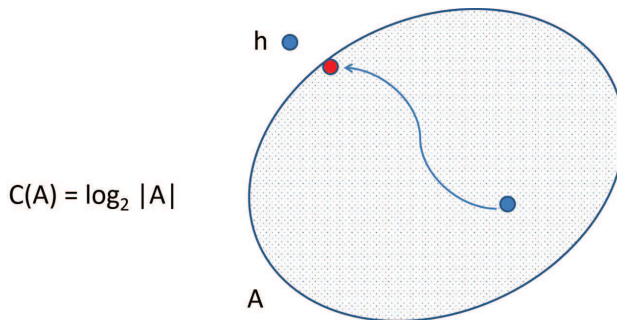

$$C(A) = \log_2 |A|$$

Figure 1: Framework.

To measure capacity in a continuous setting, one must define $|A|$ in some measure theoretic sense. Here we will simplify this problem by using Boolean neurons only, so that the class $A$ is finite and therefore we can simply define $|A|$ as the number of functions contained in $A$, as described in the next section. The capacity is also the number of bits required to specify an element of $A$.

## 2 Linear/Polynomial Threshold Functions and Notations

For theoretical and practical purposes, a neuron is often viewed as a computational unit which operates by applying a non-linear function to the dot product between the input vector and the vector of synaptic weights. Although several different non-linear operations can be considered, equating the dot product to zero defines a fundamental hyperplane that partitions the neuron's input space into two halves and provides the direction of affine hyper-planes where the dot product remains constant. Thus the most basic non-linear function that can be used is a threshold function which simply retains the sign of the dot product. When restricted to binary entries this yields the classical notion of a linear threshold gate. Thus we consider the $N$ dimensional hypercube $H = \{-1, 1\}^N$. A homogeneous linear threshold gate $f$ of $N$ variables is a Boolean function over $H$ of the form:

$$f(x_1, \ldots, x_N) = \operatorname{sign}\left(\sum_{i=1}^{N} w_i x_i\right) \tag{2}$$

Where $w = (w_i)$ is a vector of weights or parameters. Unless otherwise specified, we assume that the weights are real numbers. A non-homogeneous linear threshold gate has an additional bias $w_0$ and is given by:

$$f(x_1, \ldots, x_N) = \operatorname{sign}\left(w_0 + \sum_{i=1}^{N} w_i x_i\right) = \operatorname{sign}\left(\sum_{i=0}^{N} w_i x_i\right) \tag{3}$$

assuming that $x_0 = 1$. Throughout this paper, we exclude the cases where the activation is exactly equal to 0, as they are not relevant for the problems considered here. Linear threshold gates represent an important but very small class of Boolean functions and understanding how small, i.e. counting the number of linear threshold functions of $N$ variables is one of the most fundamental problems in the theory of neural networks.

In search for greater biological realism or more powerful computing units, it is natural to introduce polynomial, or algebraic, threshold functions by assuming a polynomial, rather than linear, integration of information in the neuron's dendritic tree. Again equating the polynomial to zero provides an algebraic variety that partitions the neuron's input space and leads to the notion of polynomial threshold gates. Thus, a homogeneous polynomial threshold gate of degree $d$ is a Boolean function over $H$ given by:

$$f(x_1, \ldots, x_N) = \operatorname{sign}\left(\sum_{I \in \mathcal{I}_d} w_I x^I\right) \tag{4}$$

where $\mathcal{I}_d$ denotes all the subsets of size $d$ of $\{1, 2, \ldots, N\}$ and if $I = (i_1, i_2, \ldots, i_d)$ then $x^I = x_{i_1} x_{i_2} \ldots x_{i_d}$, and $w = (w_I)$ is the vector of weights. Note that on $H$, for any index $i$, $x_i^2 = +1$ and therefore integer exponents greater that 1 can be ignored. Similarly, a (non-homogeneous) polynomial threshold gate of degree $d$ is given by the same expression:

$$f(x_1, \ldots, x_N) = \operatorname{sign}\left(\sum_{I \in \mathcal{I}_{\leq d}} w_I x^I\right) \tag{5}$$

the difference being that this time $\mathcal{I}_{\leq d}$ represents all possible subsets of $\{1, 2, \ldots, N\}$ of size $d$ or less, including possibly the empty set associated with a bias term. Note that for most practical purposes, including developing more complex models of synaptic integration, one is interested in fixed, relatively small values of $d$. Again, this gives rise to the fundamental problem of estimating the number of polynomial threshold gates of $N$ variables of degree $d$, with the special case above of linear threshold functions corresponding to the case $d = 1$, which will be addressed in the next section.

We use $\mathcal{A}$ to denote an architecture, i.e. a circuit of interconnected polynomial threshold functions, $T[\mathcal{A}]$ to denote the number of Boolean functions that can be implemented by $\mathcal{A}$ as the weights of the neurons are varied, and $C[\mathcal{A}] = \log_2 T[\mathcal{A}]$ to denote its capacity. We will write $\mathcal{A} = [N]$ to denote $N$ gates fully interconnected to each other (fully recurrent network), or $\mathcal{A} = [N_0, N_1, \ldots, N_L]$ to denote a layered feedforward architecture with $N_0$ inputs, $N_1$ gates in the first layer, $N_2$ gates in the second layers, and so forth. Unless otherwise specified, we assume full connectivity between each layer and the next, although most of the theory to be presented can be applied to other connectivity schemes. Much of our focus here is going to be on the special case $\mathcal{A}[N, 1]$, denoting a single neuron with $N$ inputs, since it is essential to understand first the capacity of individual building blocks. The degree of the polynomial threshold gates is either clear from the context, or specified as a subscript. We use a "*" superscript when the threshold functions are homogeneous. Thus, for instance, $C_d[N, 1]$ is the logarithm base two of the number $T_d[N, 1]$ of polynomial threshold gates of degree $d$ in $N$ variables, and $C_d^*[N, 1]$ is the same number when the gates are forced to be homogeneous.

The following relationships are straightforward to prove.

**Proposition 1:** The numbers $C_d[N, 1]$ and $C_d^*[N, 1]$ ($d = 1, \ldots, N$) satisfy the following relationships for any $N \geq 1$:

$$C_1[N, 1] = C_1^*[N + 1, 1] \tag{6}$$

$$C_{d-1}^*[N, 1] < C_d^*[N, 1] < C_1^*\left[\binom{N}{d}, 1\right] \tag{7}$$

true for any $d \geq 2$,

$$C_{d-1}[N, 1] < C_d[N, 1] < C_1\left[\binom{N}{0} + \binom{N}{1} + \ldots \binom{N}{d}, 1\right] \tag{8}$$

true for any $d \geq 2$,

$$C_d^*[N, 1] \approx C_d[N, 1] \tag{9}$$

for $d \geq 1$ and $N$ large.

## 3 The Capacity of Single Threshold Neurons

The first fundamental question is to estimate how many polynomial threshold functions in $N$ variables of degree $d$ exist. We first review the known results for $d = 1$ and then state our more general results for any fixed $d \geq 1$.

### 3.1 Linear Threshold Functions

We begin with linear threshold gates ($d = 1$). A number of well known Boolean functions are linearly separable and computable by a single linear threshold gate. For instance AND, OR, NOT are linearly separable. However, many other Boolean functions (e.g. PARITY) are not. In fact, there are $2^{2^N}$ Boolean functions of $N$ variables and the majority of them is not linearly separable. Estimating $T_1[N, 1]$ or $C_1[N, 1]$ is a fundamental problem in the theory of neural networks and it has a relatively long history [1]. The upper bound:

$$C_1[N, 1] \leq N^2 \tag{10}$$

for $N > 1$, has been known since the 1960s (e.g. [7]; see also [6]). Likewise lower bounds of the form:

$$\alpha N^2 \leq C_1[N, 1] \tag{11}$$

with $\alpha < 1$ were also derived in the 1960s. For instance, Muroga proved a lower bound of $N(N - 1)/2$ (e.g. [8]), leaving open the question of convergence and the correct value of $\alpha$. The

problem of determining the convergence and the right order was finally settled by Zuev [13, 14] who proved that:

$$C_1[N, 1] = N^2 (1 + o(1)) \tag{12}$$

Thus in short the capacity is $N^2$, as opposed to $2^N$ for the total number of functions. Intuitively, Zuev's result is easy to understand from an information theoretic point of view as it says that a linear threshold function is fully specified by providing $N^2$ bits, corresponding to $N$ examples of size $N$. These are the $N$ support vectors, i.e. the $N$ points mapped to +1 that are closest to the separating hyperplane. Conversely, it can also be interpreted as stating that $N^2$ bits can be stored in a linear threshold function as its weights are varied.

### 3.2 Polynomial Threshold Functions

For fixed $d > 1$, as well as slowly increasing values of $d$, the problem is considerably more difficult. Let us introduce the notation:

$$\binom{n}{\leq k} = \binom{n}{0} + \binom{n}{1} + \ldots \binom{n}{k} \tag{13}$$

The upper bound:

$$T_d[N, 1] \leq 2 \binom{2^N - 1}{\leq D} \quad \text{where} \quad D = \binom{N}{\leq d} \tag{14}$$

was shown in [3] (see also [1]), for any $1 \leq d \leq N$. For any $N > 1$ and $1 \leq d \leq N$, we can show that this leads to the following simple upper bound [4]:

$$C_d[N, 1] \leq \frac{N^{d+1}}{d!} \tag{15}$$

The lower bound:

$$\binom{N}{d+1} \leq C_d[N, 1] \tag{16}$$

was derived in [11]. This lower bound is approximately $N^{d+1}/(d+1)!$, which leaves a multiplicative gap $O(d)$ between the upper and lower bounds. Here we introduce the following theorem, which finally settles this gap, and contains Zuev's result as a special case:

**Theorem 3.1** *For any fixed $d$, the capacity of a polynomial threshold function of degree $d$ satisfies*

$$C_d[N, 1] = \frac{N^{d+1}}{d!} (1 + o(1)) \tag{17}$$

*as $N \to \infty$.*

The proof of this result is fairly involved ([4], as it requires generalizing the theory of random matrices to a theory of random tensors. Although we stated Theorem 3.1 for a fixed degree $d$, which is the main case of interest here, we can allow $d$ to grow mildly with $n$, and the result still holds if $d = o(\sqrt{\log N / \log \log N})$. Theorem 3.1 states that in order to specify a polynomial threshold function in $n$ variables and with degree $d$, one needs approximately $N^{d+1}/d!$ bits. This corresponds to providing the $N^d/d!$ support vectors on the hypercube that belong to the +1 class and are closest to the separating polynomial surface of degree $d$. Equivalently, Theorem 3.1 determines the complexity of a polynomial classification problem: there are approximately $2^{N^{d+1}/d!}$ different ways to separate the points of the Boolean cube $\{-1, 1\}^n$ into two classes by a polynomial surface of degree $d$ (the zero set of a polynomial).

As an aside, note that any Boolean functions of $N$ variables can be written in conjunctive normal form and thus represented by a polynomial threshold function of degree $N$. A conjecture of Aspnes *et al.* [2] and Wang-Williams [12] states that for most Boolean functions $f(x)$, the lowest degree

of $p(x)$ such that $f(x) = \text{sign}(p(x))$ is either $\lfloor N/2 \rfloor$ or $\lceil N/2 \rceil$. This conjecture was proved up to additive logarithmic terms in [9]; see also related results in [10]. This result and Theorem 3.1 both show, each in its own precise way, that low-degree polynomial threshold functions form a very small, but very important, class of all possible Boolean functions.

## 4   The Capacity of Other Neuronal Models

### 4.1   Polynomial Threshold Functions with Binary Weights.

In some situations (e.g. discrete synapses), it is useful to use models where the binary weights are bounded or even restricted to a discrete set. For instance, we can consider the binary case with weights in $\{-1, 1\}$ leading to the set of binary-weight polynomial threshold functions of degree $d$, also known as "signed majorities" in the $d = 1$ case. We are interested in estimating the number $BT(N, d)$ of such functions, or $BT^*(N, d)$ in the homogeneous case.

**Theorem 4.1** *For $d = 1$ and any $N$, the number of binary-weight linear threshold functions $BT[N, 1]$ satisfies*

$$\log_2 BT^*[N, 1] = N \quad \text{if } N \text{ is odd}$$
$$\log_2 BT[N, 1] = N + 1 \quad \text{if } N \text{ is even}$$

In short, the capacity of binary-weight linear threshold function is linear, rather than quadratic.

**Proof:** A binary-weight homogeneous linear threshold function in $N$ variables has $N$ coefficients and thus there is at most $2^N$ such functions. If $N$ is odd, consider two such functions $f_1$ and $f_2$ and assume that they differ in at least one coefficient. We want to prove that $f_1 \neq f_2$. Let $A$ be the set of indices where $f_1$ and $f_2$ have the same coefficients, and $B$ the set of indices where $f_1$ and $f_2$ have different coefficients. Obviously $|A| + |B| = N$ and $|B| \geq 1$. For any vector $x$ on the hypercube, we can write: $f_1(x) = \text{sign}\left(\sum_{i \in A} w_i x_i + \sum_{i \in B} w_i x_i\right)$ and $f_2(x) = \text{sign}\left(\sum_{i \in A} w_i x_i - \sum_{i \in B} w_i x_i\right)$. So we need to construct a vector $x$ such that $\sum_{i \in A} w_i x_i$ is as close as possible to 0, and $|\sum_{i \in B} w_i x_i|$ is as large as possible. If $|A|$ is even, then it is easy to select a vector $x$ such that $\sum_{i \in A} w_i x_i = 0$ (using alternating signs) and $|\sum_{i \in B} w_i x_i| = |B| \geq 1$ (using consistent signs) so that $f_1(x) \neq f_2(x)$. If $|A|$ is odd, then $B$ must be even and thus $B \geq 2$. It is then easy to select $x$ in a similar way such that $\sum_{i \in A} w_i x_i = 1$ and again $|\sum_{i \in B} w_i x_i| = |B| \geq 2$ so that $f_1(x) \neq f_2(x)$. The reasoning is similar in the non-homogeneous case with $N$ even. These results are exact and hold for finite $N$. $\square$

Note that in the homogeneous case, if $N$ is even then for any binary-weight homogeneous linear threshold function $f$ there are $\binom{N}{N/2}$ points on the hypercube where $f(x) = 0$ and thus $f$ is not well defined, and similarly in the non-homogeneous case when $N$ is odd. However if we extend the definition of threshold functions for instance by arbitrarily deciding that $\text{sign}(0) = +1$, then it is easy to see that for every $N$:

$$\log_2 BT^*[N, 1] = N \quad \text{and} \quad \log_2 BT[N, 1] = N + 1$$

When $d > 1$, we still have the obvious upper bounds $\log_2 BT_d[N, 1] \leq \sum_{k=1}^{d} \binom{N}{k}$ and $\log_2 BT_d^*[N, 1] \leq \binom{N+d-1}{d}$ which are true for every $N$. However it is unclear how often two different assignments of binary weights result in the same threshold function. Thus estimating $\log_2 BT_d[N, 1]$, or $\log_2 BT_d^*[N, 1]$, remains an open problem for $d > 1$.

### 4.2   Polynomial Threshold Functions with Positive Weights.

In some other situations (e.g. purely excitatory neurons), it is useful to use models where the signs of the weights are constrained. For instance, if we constrain all the weights to be positive this leads to the set of positive-weight polynomial threshold functions of degree $d$. When $d = 1$, this is a subset of the set of monotone Boolean functions. We are interested in estimating the number $PT_d[N, 1]$, or $PT_d^*[N, d]$ in the homogeneous case, of such functions.

**Theorem 4.2** *For $d = 1$ and every $N$, the number of positive-weight linear threshold functions $PT(N, 1)$ satisfies*

$$PT^*[N, 1] = T^*[N, 1]/2^N$$

*and*

$$PT^*[N,1] \leq PT[N,1] \leq PT^*[N+1,1]$$

*As a result*

$$\log_2 PT(N,1) = N^2(1+o(1))$$

In short, for $d = 1$, when the synaptic weights are forced to be positive the capacity is reduced but still quadratic.

**Proof:** The first statement results immediately from the symmetry of the problem and the fact that there are $2^N$ orthants, each corresponding to a different sign assignment to each component. The second statement is obvious. Note that the first two statements are true for any value of $N$. Finally, the last asymptotic statement is obtained by applying Zuev's result, noting that the reduction in capacity is absorbed in the $o(1)$ factor.□

For $d > 1$, the symmetry arguments breaks down. We can still write the upperbound $\log_2 PT^*[N,d] \leq \log_2 PT^*[\binom{N+d-1}{d}, d]$ but it may not be tight. Thus, in short, determining the behavior of $\log_2 PT[N,d]$ or $\log_2 PT^*[N,d]$ for $d > 1$ is an open problem.

A summary of the asymptotic results on the capacity of single neurons, stratified by degree and synaptic weight restrictions, is provided by Figure 2.

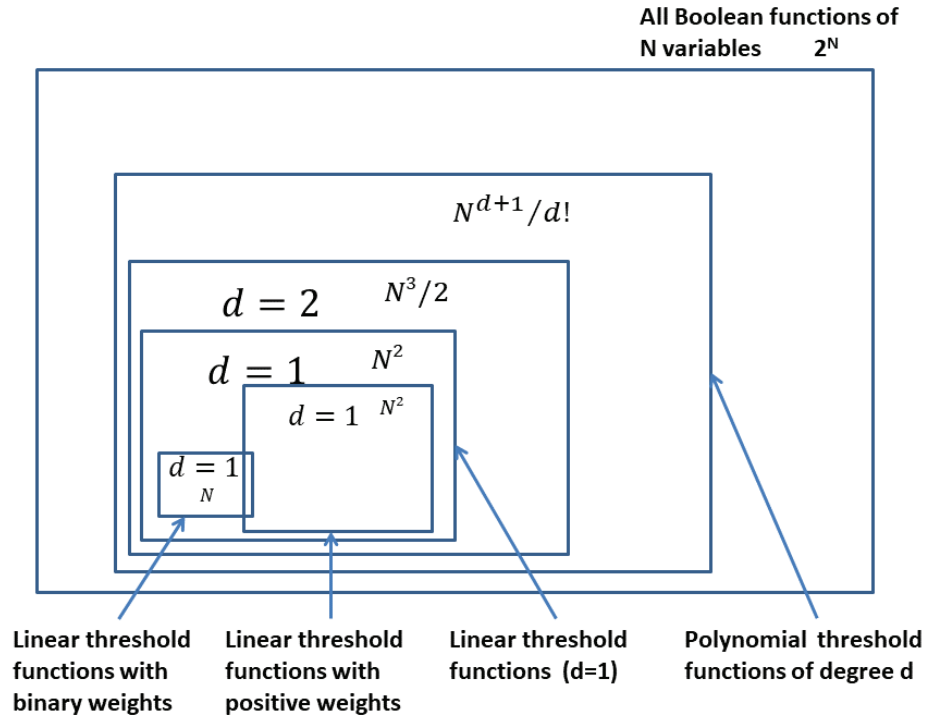

Figure 2: Stratified capacity of different classes of Boolean functions of $N$ variables. Linear threshold functions with binary weights have capacity $N$. Linear threshold functions with positive weights have capacity $N^2 - N$. Linear threshold functions have capacity $N^2$. Polynomial threshold functions of degree 2 have capacity $N^3/2$. More generally, polynomial threshold functions of degree $d$ have capacity $N^{d+1}/d!$ (fixed or slowly growing $d$). All these results are up to a multiplicative factor of $(1+o(1))$. The capacity of linear ReLU functions scales like $N^2$. The set of all Boolean functions has capacity exactly equal to $2^N$.

### 4.3 ReLU Functions

The ReLU transfer function, often used in neural networks, is defined by $f(x) = \max(0, x)$ and one can naturally define homogeneous or non-homogeneous polynomial ReLU function by letting $x$ be a homogeneous or non-homogeneous polynomial of $N$ inputs. ReLU is convenient because it is differentiable almost everywhere, its derivative is either 0 or 1, and its large dynamic range is attractive. While more powerful than simple threshold gates, our intuition is that they do not fundamentally alter the capacity of a neuron. To see this we must compute the capacity of ReLU units. However, ReLU functions are not binary and therefore cannot be compared directly with polynomial threshold gates in terms of capacity. However we can put a linear threshold gate on top of a ReLU gate to produce a binary output in order to enable comparisons (note: the same binarization approach can be applied to *networks* of ReLU or other real-valued gates).

To fix the ideas, let us consider the case of greatest interest corresponding to $d = 1$ (but the same ideas apply to $d > 1$). We can then consider two $\mathcal{A}[N, 1, 1]$ architectures, one comprised of two linear threshold gates and one comprised of a ReLU gate followed by a linear threshold gate. Let $C[N, 1, 1]$ be the capacity of the first one, and $C_{ReLU}[N, 1, 1]$ the capacity of the second architecture with the ReLU function. To limit the contribution of the top gate we force its main weight to be equal to +1 but its bias to be arbitrary (this is necessary also to avoid cases where the input to the top threshold gate is equal to 0). In other words, if the lower gate as weights $w_i$ and activation $S = \sum_{i=0}^{N} w_i x_i$, then the final output is given by $O = \text{sign}(\text{sign}\, S + b)$ in the pure threshold gate case, and by $O = \text{sign}(ReLU(S) + b)$ in the ReLU case. Under these conditions, we have the result:

**Theorem 4.3**

$$C[N, 1, 1] = C[N, 1] = N^2(1 + o(1)) \tag{18}$$

$$C_{ReLU}[N, 1, 1] = C[N, 1] + N - 1 = N^2(1 + o(1)) \tag{19}$$

**Proof (Sketch):** For the architecture containing only threshold gates, the output gate can only implement one of three functions: Identity, TRUE (always +1), FALSE (always -1). All these functions can be incorporated directly into the threshold gate of the hidden layer, and thus they do not increase (or decrease) the number of functions that can be computer by the threshold gate of the hidden layer, which has capacity $N^2(1 + o(1))$ by Zuev's result, and thus we have Equation 18. In the case of the ReLU gate, if $b > 0$ then the final output is always equal to +1, which corresponds to only one function. The only interesting case is for values of $b < 0$. For each $b < 0$ the overall function is a new linear threshold gate associated with a translated hyperplane $S + b = \sum_i w_i x_i + b = 0$. Note that letting $b$ vary is essentially to utilize the additional power that ReLU function have in their linear regime. Thus as $b$ is varied, the hyperplane $S = 0$ is translated by different amount and every time it crosses a corner of the hypercube a new Boolean function is being created. This happens at most $2^N$ times, and typically $2^{N-1}$ times, since $b$ must be negative and translations occur only in one direction. Thus the lower layer implements on the order of $2^{N^2}$ different functions, or hyperplanes, and on average each one of them gives rise to $2^{N-1}$ functions, for a total of approximately $2^{N^2+N-1}$ functions, which leads to Equation 19. In short, for $d = 1$, when a ReLU transfer function is used the capacity is increased but remains quadratic. $\square$

## 5 General Bounds for Networks

While interesting, the previous results apply to single neurons and of course we are interested in networks containing many interconnected neurons. For these cases, the general strategy is to first get upper bounds and lower bounds, and then check whether any gap between the lower and upper bound can be reduced. In general, one always has the simple upperbound:

$$C[network] \leq \sum_i C[neuron_i] \tag{20}$$

In other words, the total capacity of a newtowrk is always upperbounded by the sum of the capacities of all the individual neurons (this remains true even when the circuit contains threshold gates of different degrees and different fan-ins).

## 5.1 Fully Connected Recurrent Networks

In the case of a fully interconnected recurrent network of threshold functions of degree $d$, the number $\mathcal{A}_d[N]$ of functions that can be implemented is obviously bounded by:

$$T_d[N] \le (T_d[N,1])^N \le 2^{\frac{N^{d+2}}{d!}} \tag{21}$$

since we can choose a different threshold gate for each node of the network. In the case of a fully connected network, in principle one must further define how the gates are updated (e.g. stochastically, synchronously) and what defines the function computed by the network for each set of initial conditions (e.g. sequence of states versus limit when it exists). Regardless of the mode of update, we will use the definition that two $\mathcal{A}_d[N]$ architectures with different weights compute the same function if and only if for any set of initial conditions they produce the same sequence of states under the same update scheme (note: this require the units to be numbered 1 to $N$ in each network). Under this definition, it is easy to see that the upperbound above becomes also a lower bound and thus one has the theorem:

**Theorem 5.1** *For $N$ large enough, the capacity of a fully connected network of $N$ polynomial threshold gates of degree $d$ is given by:*

$$C[N] = \frac{N^{d+2}}{d!}(1 + o(1)) \tag{22}$$

In short, in the main case where $d = 1$, this states that the capacity is a cubic function of the number of neurons. While this result is useful, it corresponds to an architecture that is amorphous. The cases of greatest interest for deep learning applications are cases where there are constraints on the connectivity, for instance in the form of a layered feedforward architecture.

## 5.2 Layered Feedforward Architecture

To illustrate how the techniques developed so far can be applied to feedforward layered architectures, consider a $\mathcal{A}_d[N, M, 1]$ architecture. We have the theorem:

**Theorem 5.2**

$$C_d[N,1] \le C_d[N,M,1] \le MC_d[N,1] + C_d[M,1] \le M\frac{N^{d+1}}{d!} + \frac{M^{d+1}}{d!} \tag{23}$$

**Proof:** The lower bound is provided by the capacity of a single unit, and the upper bound is again the sum of all the capacities. $\square$

If we take $d = 1$, this gives a weak lower bound that scales like $N^2$ and an upper bound that scales like $MN^2 + M^2$. When $M$ is small with respect to $N$, the upper bound scales like $MN^2$. When $M$ is large with respect to $N$, the output gate is still limited by the fact that it can have at most $2^N$ distinct inputs, as opposed to $2^M$. Thus in fact one can prove that the upper bound scales like $MN^2$ in all cases. Furthermore, through a constructive proof, this is also true for the best lower bound. More precisely, we show in [5] that in general there exist two constants $c_1$ and $c_2$ such that $c_1 MN^2 \le C_1[N, M, 1] \le c_2 MN^2$, and $C[N, M, 1] = MN^2(1 + o(1))$ if $N \to \infty$ and $\log_2 M = o(N)$. In addition, we also show how generalize this result and derive tight bounds on the capacity of general $\mathcal{A}[N_0, \ldots, N_L = 1]$ feedforward architectures in terms of the quantity $\sum_{k=0}^{L-1} \min(N1, N_2, \ldots, N_k)N_k N_{K+1}$. He we sketch the proof for the single hidden-layer case.

**Theorem 5.3** *The capacity of an $A[N, M, 1]$ architecture of threshold gates satisfies:*

$$C[N, M, 1] = MN^2(1 + o(1)) \tag{24}$$

*for $N \to \infty$ and for any choice of $M \in [1, 2^{o(N)}]$.*

**Proof (Sketch):** Let us denote by $f$ the map between the input layer and the hidden layer and by $\phi$ the map from the hidden layer to the output layer. For the upper bound, we first note that the total number of possible maps $f$ is bounded by $2^{MN^2(1+o(1))}$, since $f$ consists of $M$ threshold gates, and each threshold gates correspond to $2^{N^2(1+o(1))}$ possibilities by Zuev's theorem. Any fixed map $f$, produces at most $2N$ distinct vectors in the hidden layer. It is known [1] that the number of threshold functions $\phi$ of $M$ variables defined on at most $2^N$ points is bounded by:

$$2\binom{2^N - 1}{\leq M} = 2^{NM(1+o(1))} \tag{25}$$

using the assumption $M \leq 2^{o(N)}$. Thus, under our assumptions, the total number of functions of the form $\phi \circ f$ is bounded by the product of the bounds above which yields immediately:

$$C[N, M, 1] \leq MN^2 \left(1 + o(1)\right) \tag{26}$$

To prove the lower bound, we use a procedure we call filtering. For this, we decompose $N$ as: $N = N^- + N^+$ where $N^- = \lceil \log_2 M \rceil$. Likewise, we decompose each input vector $I = (I_1, \ldots, I_N) \in \{-1, +1\}^N$ as: $I = (I^-, I^+)$, where:

$$I^- = (I_1, \ldots, I_{N^-}) \in \{-1, +1\}^{N^-} \quad \text{and} \quad I^+ = (I_{N^-+1}1, \ldots, I_N) \in \{-1, +1\}^{N^+} \tag{27}$$

For any Boolean linear threshold map $f^+$ from $\{-1, +1\}^{N^+}$ to $\{-1, +1\}^M$, we can uniquely derive a map $f = (f_1, \ldots, f_M)$ from $\{-1, +1\}^N$ to $\{-1, +1\}^M$ defined by:

$$f_i(I^-, I^+) = [I^- = i] \ \ AND \ \ [f_i^+(I^+)] \tag{28}$$

Here $I^- = i$ signifies that the binary vector $I^-$ represents the digit $i$. In other words $I^- = i$ is used to select the $i$-th unit in the hidden layer, and filter $f^+$ by retaining only the value of $f_i^+$. It can be checked that this selection procedure can be expressed using a single threshold function of the input $I$. We say that $f$ is obtained from $f^+$ by filtering and $f$ is a threshold map. It is easy to see that the filtering of two distinct maps $f^+$ and $g^+$ results into two distinct maps $f$ and $g$. Now let us use $\phi = OR$ in the top layer–note that OR can be expressed as a linear threshold function. Then it is also easy to see that $\phi \circ f \neq \phi \circ g$. Thus the total number of Boolean functions that can be implemented using linear threshold gates in the $A[N, M, 1]$ architecture is lower bounder by the number of all Boolean maps $f^+$. This yields:

$$C[N, M, 1] \geq M(N^+)^2 \left(1 + o(1)\right) = MN^2 \left(1 + o(1)\right) \tag{29}$$

using the fact that $N^+ = N - \lceil \log_2 M \rceil$, and $\lceil \log_2 M \rceil = o(N)$ by assumption. $\square$

# 6   Conclusion

The capacity of a finite class of functions can be defined as the logarithm of the number of functions in the class. For neuronal models, we have shown that the capacity is typically a polynomial in the relevant variables. We have computed this polynomial for individual units and for some networks. For individual units, we have computed the capacity of polynomial threshold functions of degree $d$, as well as for models with constrained weights or ReLU transfer functions. For networks, we have estimated the capacity of fully recurrent networks of polynomial threshold units of any fixed degree $d$, and have derived bounds for layered feedforward networks of polynomial threshold units of any fixed degree $d$. The notion of capacity is also connected to other notions of complexity including the VC dimension, the growth function, the Rademacher and Gaussian complexity, the metric entropy, and the minimum description length (MDL). For example, if the function $h$ to be learnt as MDL $D$, and the neural architecture being used as capacity $C < D$ then it is easy to see that: (1) $h$ cannot be learnt without errors; and (2) the number $E$ of errors made by the best approximating function implementable by the architecture must satisfy $E > (D - C)/N$. These connections will be described elsewhere.

**Acknowledgments**

Work in part supported by grants NSF 1839429 and DARPA D17AP00002 to PB, and AFOSR FA9550-18-1-0031 to RV.

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
