[Reviews · NeurIPS 2018]

Reviewer 1



Summary: The authors provide a collection of results regarding the capacity of different polynomial threshold gates of N variables with d-degree. The key result is the closing of a multiplicative gap between upper and lower bounds on the capacity of polynomial threshold functions (e.g. Thm. 3.1). They leverage these ideas to provide capacity bounds for neural network architectures. Positive: The manuscript is exceptionally clear, thorough, and of obvious practical relevance. The new lower bound proof closes an old gap in our understanding of capacities of polynomial functions. The techniques in the appendix on random tensors are also clearly powerful, and, if the open questions posed in section 10 aren’t already a good indication, this work opens many avenues for further research. Negative: While this paper is excellent, as a bit of sociological commentary, the lengths of supplementary materials for NIPS submissions are starting to get…out of hand. I understand the author’s likely desire to broadcast this work to the machine learning community, but this really might’ve been better served as a journal article, both because the review process would be of higher quality, and because the full meat of the work could all be in one place (instead of shoving a ton of really interesting stuff into Ch. 10 of the supplementary material (!!!)). Regardless, I, of course, strongly suggest acceptance publishing in NIPS. Here are some minor typos that I spotted: 6: feedfoard -> feedforward 55: synpatic -> synaptic 64: fedforward -> feedforward 85: computatble -> computable Eq. 10: leq -> \leq Supplementary material: pg. 8: betterunderstood -> better understood 3.41: arrangments -> arrangements 10.4: Poission -> Poisson (unless there’s a Poission I don’t know about…)

Reviewer 2



I would like to apologize to the authors for my dismissive original review that was based on a careless reading of the paper. This paper gives a new, tighter asymptotic bound on the number of polynomial threshold functions of n variables and degree d, closing a multiplicative gap of O(d) between the best previously known lower and upper bounds. The paper uses an impressive mathematical repertoire to solve a long-standing open problem that has close connections with learning theory. The paper's exposition could be significantly improve. I would suggest that the authors emphasize the main result of the paper (Theorem 3.1). As the paper is currently written, the main result is only introduced and discussed halfway through the paper, with the abstract, the introduction, and the conclusion giving very little information about the significance of the results. This can lead a cursory reader (such as, unfortunately, myself) to entirely miss the main contribution of the paper. I find the introductory part of the supplementary material much more informative than the paper itself -- replacing the entire main paper with the first pages of the supplementary material would seem a clear improvement in the exposition.

Reviewer 3



The main result of the paper is giving tight bounds on the number of n-variable boolean functions that can be expressed as degree d-polynomial threshold functions for any fixed d (d growing very mildly with n) also works. To me the rest of the results while interesting seem to be mostly easy applications of the key result to other more fashionable neural network models. However, the tightness of the main result does not translate to tight bounds when other neuroidal models are considered, because of the kind of non-linearities or weight constraints involved. The main result is highly non-trivial, the proof quite lengthy though elegant, and resolves a 25 year old open problem. Although the proof uses a lot of heavy mathematics, the key contribution seems to be generalizing random matrix theory to a random tensor theory -- the key result being that a large number of stochastically independent random vectors in low-dimension (and hence clearly not linearly indpendent) still yield a high degree of linear independence when tensorized. Although the kind of techniques and results may be unfamiliar to a NIPS audience, I think it would be interesting to the NIPS audience and suggest the paper be accepted. Below I've included some typos that I found in the 8-page version as well as the longer supplementary (however, the page numbers in the supplementary refer to the arxiv v1 of the paper): 8-page paper: Line 23: contained in |A -> remove the | Line 40: class of Boolean function -> functionS Line 59: Section -> section Line 60: W -> We Eq(7) : Missing [] in the middle term Eq(10) : leq => \leq I'm not really sure how to interpret the various results Sec 4.2 onwards that read of the form N^2(1 + o(1)) +/- \Theta(N) terms. Unless we know that o(1) is actually o(1/N), the additional \Theta(N) terms are essentially meaningless? Arxiv version: Page 21 (towards end of Proof lemma 4.4): (Cauchy-Schwarz step) -> the equality should be an inequality Page 22 (middle of page just below equation) : extreme combinatorics -> extremal combinatorics? Page 25: I believe there are some typos in Lemma 5.3, e.g. V^\perp - U^\perp may not even be a vector space, so I'm not sure what its dimension would be. There appear to be a couple of typos in the proof as well. Page 27: Third (text) line of Proof of Lemma 5.6 : \theta^\perp -> \theta^\top Page 28: In Sec 5.4 sometimes E_x and Ex are used to mean the same thing

Reviewer 4



Overall assessment: I consider this work to be interesting and worth accepting, even though I am not convinced by the presentation itself, which I feel is obfuscating the real point of the paper. (More details to follow.) ----------------------------------------------------------------------------- Details: In this work, the author(s) consider, under a specific angle, the following quite fundamental question: "what is the expressive power of a given class C of functions (concepts)?" A natural (though maybe arguable, and not suitable for *all* purposes) is to consider the logarithm of its size, log_2 |C| (or, if C is infinite, the log of its measure, for a well-chosen notion thereof; or the size of a minimum cover). Indeed, this is an approach well-studied in information theory and statistics, for instance (see the related notions of VC dimension, covering number, bracketing entropy, etc.). With this in mind, the paper seeks to obtain tights bounds for classes C of Boolean functions (f: {0,1}^n \to {0,1}) such as (i) linear threshold functions (LTF, halfspaces) (ii) polynomial threshold functions (PTF), (iii) restrictions thereof (small weights, positive weights), and (iv) networks obtained by combining them. Now, this is indeed interesting. My main point is that these questions are quite hard, and as far as I can tell the results for (iii), and (iv) are quite weak or straightforward. The main contribution of the paper is Theorem 3.1, which provides and actual asymptotically tight bound for (ii), the class of degree-d PTFs (concluding a decades-long line of work, the previous bounds still being off by a factor d). More specifically: - Theorem 1 is a preliminary, and should be either a fact or a lemma - Theorem 3.1 is the main technical result, non-trivial, and (in my opinion) should be highlighted in the introduction as the key result - Theorem 4.1 is quite straightforward and easy; and only deals with (binary-weights) LTFs, not PTFs - Theorem 4.2 is not tight; and only deals with (positive-weights) LTFs, not PTFs - I am not able to really quantify the relevance of Theorems 5.1 and 5.2. In summary, again: I feel the presentation distracts from the main result (Theorem 3.1), promising instead results about more general classes (degree-d PTFs, networks of PTFs and LTFs, more general types of functions...) which end up being unaddressed or for which the results are much less impressive (as in, they are not tight, or are simple corollaries). This is a bit disappointing, since Theorem 3.1 is very good by itself. (Ironically, this is exactly what the supplemental material is: a math paper focusing entirely on, and dedicated to, Theorem 3.1) ----------------------------------------------------------------------------- More detailed comments (technical, and typos) below. - I am really not convinced Figure 1 is useful in any way to understand the paper. - l.41 Misleading. Phrased as it is, it sounds like this problem is still open, while (as you mention later) it has been settled since '89. - l.60 and everywhere: "i.e. " -> "i.e., " - eq (10) (l.89): "leq" -> "\leq" - Section 3.1 (e.g., l.99): you should also discuss Chow's theorem, which gives such an interpretation and upper bound, as it states that any LTF is uniquely determined by its n+1 first Fourier coefficients. Now, since for a Boolean function each Fourier coefficient as granularity 1/2^{n-1}... (see e.g. discussion after Theorem 2.2 in https://arxiv.org/abs/1511.07860) - l.104: similar: you should discuss the degree-d PTF extension of Chow's theorem, due to Bruck [1990]: see e.g. discussion at the end of https://arxiv.org/abs/1206.0985 - Section 4.1: for the sake of the literature review, mention that binary-weight LTFs are commonly referred to as "signed majorities". Also, this section is quite misleading. You start by discussion binary-weight PTFs (degree d, for general d), but prove essentially nothing (except a possibly very loose easy upper bound) for d>1; only considering d=1 (Theorem 4.1). Moreover, a really relevant and important case (as you mention!) is the case of bounded weights, say when the weights are constrained to be poly(n) (see e.g. https://arxiv.org/abs/1508.03061 (STOC'17) and references within). Can you say anything for that case? - l.147: trailing period. - Theorem 4.2: where was T[N,1] introduced? I couldn't find that notation -- it looks like you are using it in lieu of C_1[N,1]? Moreover, I don't buy the "As a result" statement (and the "reduced by a linear factor" which follows). You have N^2(1+o(1)), so that the second order term you neglect is o(N^2). Since N is also o(N^2), you *cannot* write and discuss this -N term! It could be completely cancelled or drowned by the o(N^2) (which, as far as the statement of the theorem promises, cound be anything, say N^{1.999999}) - ll.178-179: why is it a reasonable thing to do? Detail. - Eq. (17) "\approx =" -> "\approx" Also, more generally (Eq. 16, Eq. 19, Theorem 5...), you never have defined what \approx means for you. Is it informal? Is it notation for the Landau asymptotic \sim? You need to explain, as otherwise one cannot make sense of it. For instance, in (17) using the Landau asymptotic equivalence (which only "guarantees" anything meaningful for the first dominating term) would be strange, as you give 3 terms. Do you mean instead "= N^2 +N + o(N)"? "= N^2 +N + O(1)"? "= N^2 +N - 1 +o(1)" (which given the fact that sizes are integers, would lead to an actual exact equality...)? In short: you are abusing asymptotic notation here, giving too many terms (leading to a mistaken feeling of precision) l.210: typo ("threshold") l.236: typo ("layered") l.272: typo (ref [17]): "O'Donnell" Again, the section title and introduction promise results about degree-d PTFs, but the only actual result is for d=1.